TECHNICAL RELEASE

# Biodepot Launcher: an app to install, manage and launch bioinformatics workflows

Ling-Hong Hung[1,†], Thomas J. Dahlstrom[1,†], Johnalbert Garnica[1], Emmanuel Munoz[2], Robert Schmitz[2] and Ka Yee Yeung[1,*]

1 School of Engineering and Technology, University of Washington Tacoma, Tacoma, WA, USA
2 Biodepot LLC, Seattle, WA, USA

## ABSTRACT

We present the Biodepot Launcher, a desktop application that facilitates installation, management and deployment of bioinformatics workflows using the Biodepot-workflow-builder (Bwb). With the new app, Bwb can be started by double-clicking on an icon, eliminating the need for typing cryptic start up commands into the terminal. This creates an end-to-end graphical and easy-to-use interface to manage and launch containerized workflows on the local computer or cloud instances. Biodepot Launcher is written in React and Javascript, and uses the node.js framework Neutralinojs and web browser routines to allow the application to execute on Linux, Windows and Mac desktop environments.

**Subjects** Software and Workflows, Bioinformatics, Biomedical Science

**Submitted:** 12 September 2024

* Corresponding author. E-mail: kayee@uw.edu

† Contributed equally.

Preprint submitted at https://doi.org/10.20944/preprints202501.0562.v1

## BACKGROUND

The Biodepot-workflow-builder (Bwb) is an open-source, containerized desktop that facilitates the creation, customization and reproducible execution of bioinformatics workflows [1, 2]. Graphical *widgets* control Docker containers executing a modular task. Widgets are linked graphically with a drag-and-drop user interface to build bioinformatics workflows that can be reproducibly deployed and executed on different local and cloud platforms. Each widget contains a form-based user interface to facilitate parameter entry, and a console to display intermediate results. Many workflows have been developed for various biomedical big data in the Bwb, such as RNA-seq, DNA sequencing [3], nanopore sequencing [4, 5], and bioimage analysis [6]. While the Bwb desktop environment is very user-friendly, the launching of Bwb requires entering commands into a terminal. Management of workflows and maintenance of current versions must be manually handled by the user.

In this paper, we introduce the **Biodepot Launcher**, an open-source desktop application that facilitates the installation, management and deployment of bioinformatics workflows in the Bwb. With the launcher, we now have an end-to-end graphical application that provides a reproducible graphical desktop to create workflows to analyze biomedical data. Users can choose to launch Bwb on local or cloud infrastructure. The Biodepot Launcher app is written using React and Javascript, using nodejs through the neutralinojs framework and browser routines to render the graphics on different operating systems. In addition, the new launcher app includes tools that simplify the installation and management of workflows, deployment of Bwb locally, on GitPod and the cloud. Demo videos and training

materials are available at https://biodepot.github.io/training/running_bwb/bwb_launcher/. Biodepot Launcher does not completely eliminate the need for manual installation of dependencies. Users still need to install Docker, as all workflows and part of the Biodepot Launcher is containerized. For execution on AWS (Amazon Web Services), an AWS account, the AWS CLI (Command Line Interface) and local credentials directory are required. In addition, Docker Machine must be installed (using our provided scripts) if the user wishes to be able to terminate AWS instances through the Launcher interface. This is a very minimal set of requirements, made possible by the fact that Bwb, and its workflows and some components of Biodepot Launcher do not require explicit installation as they are implemented by downloading and executing images from our DockerHub repository.

## METHODS

Figure 1 shows a summary of the features and components in Biodepot Launcher. Biodepot Launcher takes advantage of online repositories and resources to document, store, and launch workflows in Bwb. Because workflows and documentation are stored on GitHub, updates will be readily available as changes are made.

### Launching Bwb and managing workflows

The Bwb workflow engine is a containerized desktop that does not need to be installed. The user enters a Docker command to automatically pull the container, start it and optionally load a starting workflow. Biodepot Launcher app eliminates this command line step and provides a graphical interface to start Bwb locally on a laptop or desktop compuer or on an AWS or GitPod cloud instance. We include GitPod [7] as it provides a free-tier service that can execute jobs on their cloud servers simply by clicking on a GitHub badge. We use it extensively in our training tutorials. In addition to graphically controlling the initial startup of Bwb, the Biodepot Launcher also checks, updates, downloads and organizes workflows that we maintain on GitHub [8]. Previously, users would have to manually perform these operations.

### Implementation

An increasingly common approach to implementing graphical desktop apps for different operating systems is to leverage browsers such as Chrome and Firefox for graphical support. Browsers are responsible for rendering Javascript, HTML and CSS in different environments. When coupled with a framework that bundles a browser (electronjs) or browser routines (neutralinojs) with backend support through node.js, portable graphical desktop apps can be built using the same technologies that are used to build web applications.

#### *Reactjs*

Biodepot Launcher is built using React.js [9] coupled with the Neutralinojs [10] framework. React.js is a widely used front-end development Javascript framework that enables the development of highly interactive, responsive, web applications. It is used to build the user interface for Biodepot Launcher in the same way that it would be used to develop the front end in a pure web app.

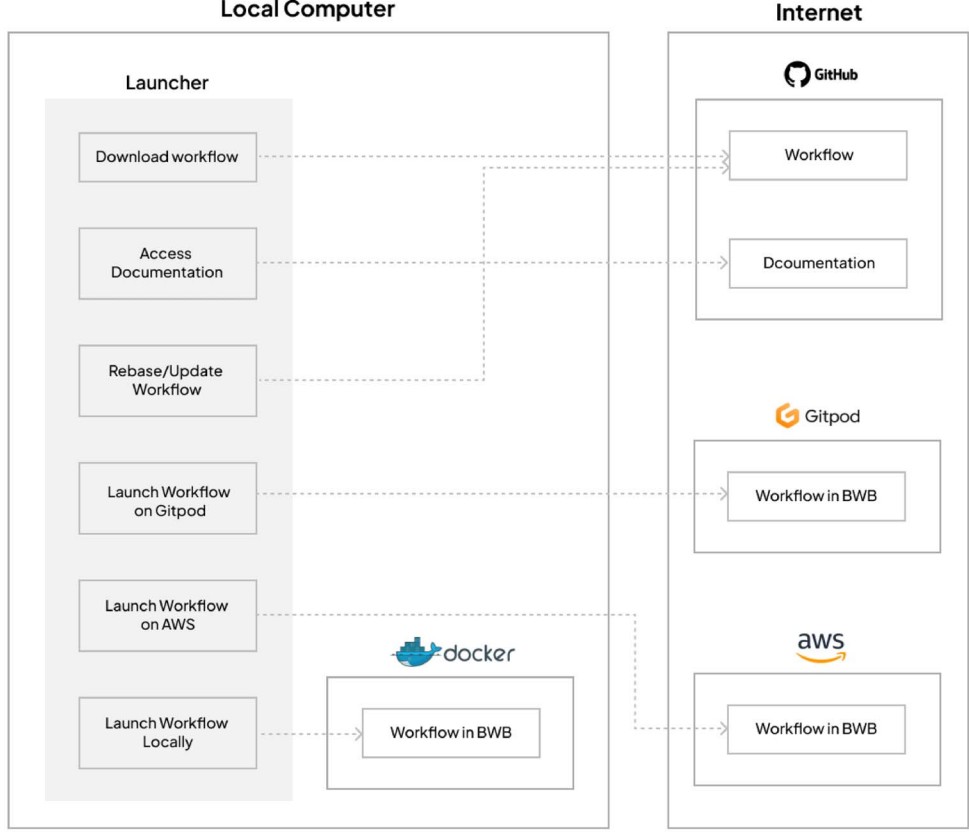

**Figure 1.** Docker needs to be installed to run Bwb. A Docker start command is executed that pulls the Bwb container, starts the workflow desktop environment and optionally loads a starting workflow. Biodepot Launcher uses a graphical interface to construct and execute the startup command to simplify the launching procedure. For cloud execution, the entire process of choosing an instance, starting the instance and starting Bwb is accomplished using Docker Machine. In addition to launching on the AWS cloud, we also support GitPod which has a free-tier that is useful for short demos and tutorials . GitHub is used to warehouse the Bwb workflows and accompanying documentation. GitHub actions is used to maintain an index that helps Biodepot Launcher manage and update workflows.

### Neutralinojs

In a web app, there is also a server backend. Node.js allows for the backend to be written in Javascript. The server can execute locally, on the same machine as the front-end. In this situation, the front-end, backend and browser can essentially function as a single graphical desktop app. Neutralinojs is an application development framework that bundles these three components to create desktop applications. Neutralinojs creates a backend server using node.js and uses the system's preferred web-browser to render a user interface. Through the backend, neutralinojs provides access to the local filesystem, networking, and other resources typical of a desktop application. These functions are normally unavailable to the front-end for security reasons. With neutralionjs we can create desktop application that has the portability and expressiveness of a web front-end while having the expanded functionality of system level commands that are not normally available to a browser. Neutralinojs compiles standalone executables for a multitude of operating system and platforms including Windows, Linux, and macOS. In contrast to electronjs, which creates stand-alone apps by bundling the entire browser with the app, neutralinojs creates much

smaller apps by bundling only the essential routines and relying on the the system's WebView capabilities.

### GitHub actions

GitHub Actions is a CI/CD system provided by GitHub. In the context of Biodepot Launcher, we use GitHub actions to create and update hashes for the Bwb workflows that we maintain. This allows us to determine whether the user downloaded workflows have changed and whether there is an update available. The GitHub Actions configuration for Biodepot Launcher creates a set of files that inform Biodepot Launcher about the workflows' file details in the GitHub repository, without the use of the GitHub API. The generated files.txt contains all the file locations in the GitHub repository, as well as if the file is a tree or blob. The second file generated is hash.txt. Each line in hash.txt is the hash from the files contained in a workflow folder and the workflow name. The method of generating the hash of a workflow is documented in the hash.sh file in the ./launcher-utils/amd64 directory of the Biodepot Launcher GitHub repository.

### Updating and launching workflows

Bwb and all Bwb workflows are containerized in order to ensure reproducibility across operating systems. These are stored in the Biodepot DockerHub and downloaded as needed. The same strategy is utilized for the launcher-utils module of the Biodepot Launcher to reproducibly provide functions that are difficult to implement directly in React and neutralino. The launcher-utils module provides two key functionalities. One is to generate a consistent hash for a workflow directory which is needed for Biodepot Launcher to determine whether there are any changed files and hence an available update. This is accomplished using the tar utility to stream the file contents in a reproducible manner to the sha256 hashing function and create a consistent hash regardless of platform. The other major function of launcher-utils is to start Amazon EC2 instances in the cloud with the Bwb desktop and workflow. This is accomplished using Docker Machine (GitLab's fork of Docker Machine [11] and Rancher Labs' Rancher Machine [12]) which is a simple mechanism for remote execution of docker commands such as the Bwb startup command. Unfortunately, due to security limitations, the dockerized version of Docker Machine is unable to stop and terminate EC2 images. The local installation of Docker Machine is invoked by Neutralinojs to perform these functions.

## Testing across operating systems

Biodepot Launcher was tested on multiple operating systems using 13 test cases. These are detailed in the test plan https://github.com/BioDepot/biodepot-launcher/blob/main/docs/Launcher_Test_Plan.pdf. The following operating systems were tested: Ubuntu, Windows 10/11 WSL2 Ubuntu, macOS (M-Series).

## INSTALLATION OF THE BIODEPOT LAUNCHER APPLICATION

To run Biodepot Launcher, use the following steps:

- Download binaries.zip from the "Project home page" link in the "Availability of source code and requirements" section.

- Unzip the binaries.zip file on your machine. Inside the unzipped folder, there will be binaries following a naming convention that ends with the OS and architecture of your CPU.
- Once the binary is selected and unzipped, the binary can be run with a simple double-click.
- For macOS: When opening the binary for the first time, there will be a security prompt. While the prompt is showing, navigate to the Security and Privacy menu in the System Settings and click the "Open Anyway" button for the application. This allows the binary to be opened.

Biodepot Launcher has a few remaining dependencies that must be installed by the user, depending on whether it is used for local or AWS execution. Docker [13] is required for local execution of Bwb, Bwb workflows and for Bwb launcher-utils. For cloud execution the AWS CLI [14], an AWS account with proper credentials (i.e. a local credentials directory) is required. Docker Machine is also required but is provided in the bundled executable. Note that the Ubuntu version requires that the user logout and log back in for Docker Machine to be accessible from anywhere on the command line. While this procedure is more complicated that we would like, it is still significantly simpler than the usual cloud setup needed for execution of complicated bioinformatics workflows.

## USAGE

### Starting screen

Figure 2 shows a screenshot of the interface when Biodepot Launcher starts. Biodepot Launcher does not automatically detect changes made to workflows in the Bwb. A reload button is included to refresh the interface in case locally installed workflows get out of sync with the Biodepot Launcher.

### Installing workflows

The "Workflow repository" button, when clicked, will show the workflows that are available for download. This button is automatically selected when starting Biodepot Launcher. The workflows that are shown in the main pane are listed by categories. Categories of workflows included in this initial release include DNA-sequencing (DNA), RNA-sequencing (RNA), and general. Examples of workflows include the Genomics Data Commons DNA-sequencing workflow [15], a simple RNA-sequencing workflow using Salmon [16], the Genomics Data Commons mRNA-sequencing workflow [17], and a bioimage workflow for focal adhesion [6]. These example workflows can be run on a GitPod instance in under half an hour. We expect to expand the list of included workflows in future releases. To download workflows from these categories, click on the category button in the main pane when the "Workflow repository" button is selected. For each workflow, its name is listed and the option to install the workflow by clicking on the "Install Now" button. Clicking the "Install Now" button will activate a download of the workflow to the computer that you launched Biodepot Launcher from. Once the workflow has finished downloading, the text in the button will change to "Installed" and the once yellow dot in the button will change to green.

When users want to use installed workflows, the "Installed Workflows" buttons can be selected. First select the category of workflow list as one of the "Installed Workflows" buttons. Once a type is selected, the button will have a purple background, and you will see

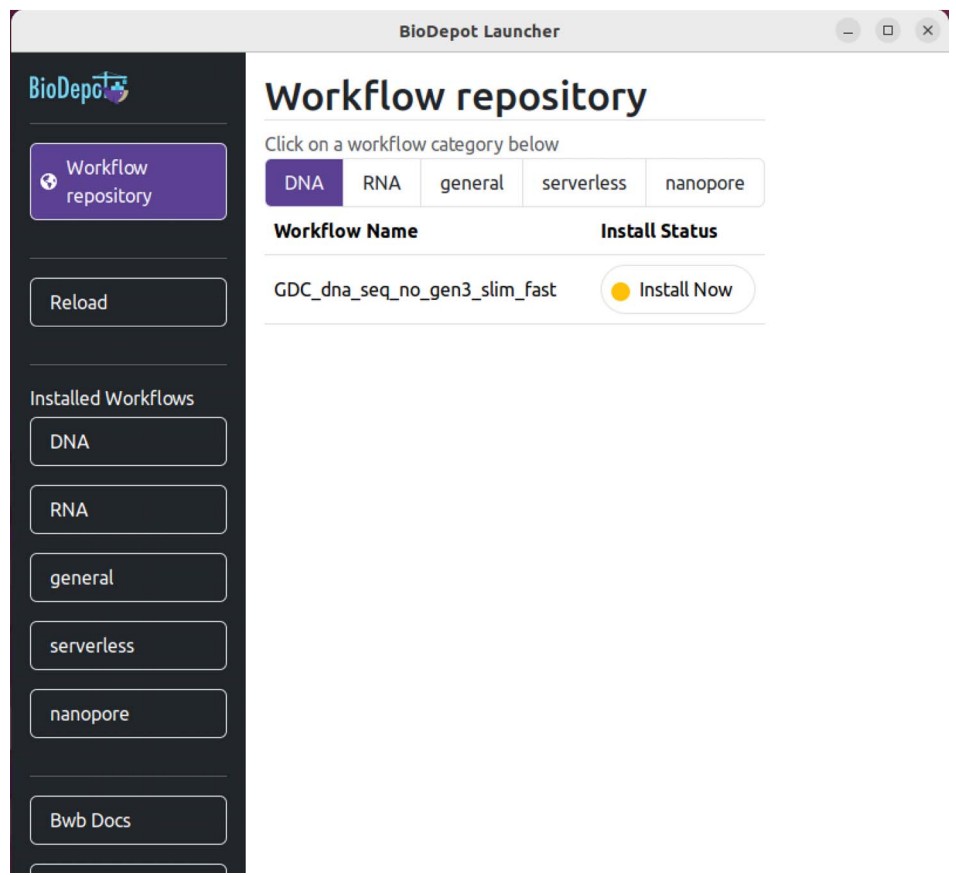

**Figure 2.** Screenshot of when Biodepot Launcher starts. The sidebar on the left includes the "Workflow repository" button which allows a user to select from the workflow category in the main pane, and subsequently any number of workflows. When clicked, the "Install Now" button will install the associated workflow on the user's machine. The sidebar includes a "Reload" button that is used when a previously installed workflow becomes out of sync. The "Installed Workflows" category lists the workflow categories of which workflows are installed under. Selecting a category will show the installed workflows of that category. The self-explanatory "Bwb Docs" button. Additionally, a "GitPod Documentation" button (which is offscreen in this figure) is the final button, which brings the user to details about how to use GitPod with the launched workflows.

your previously downloaded workflows in the main pane labeled "Installed". See Figure 3 for a screenshot of the "Installed workflows" screen.

After a workflow is installed, the workflow name will be listed with a colored dot as an indicator. The colored dot is green if the locally installed workflow is current with the one on GitHub. If the local workflow and the workflow on GitHub are not the same, the indicator dot will turn yellow and the text of the button next to it will state "Update Available". Selecting the "Update Available" button will start a process that will update the local workflow to the more current one on GitHub. If the locally installed workflow has changes made by the user, the indicator dot will remain green and a "Rebase" button will be beside the indicator. Selecting the "Rebase" button will roll back all changes made to the workflow since the last install/update. In both cases all changes made by the user will be lost and there will be a prompt indicating this before the user makes a final decision.

In addition, there will be two icons present for each workflow listed. The first icon is a book that indicates if documentation is present for the workflow on GitHub. Clicking this

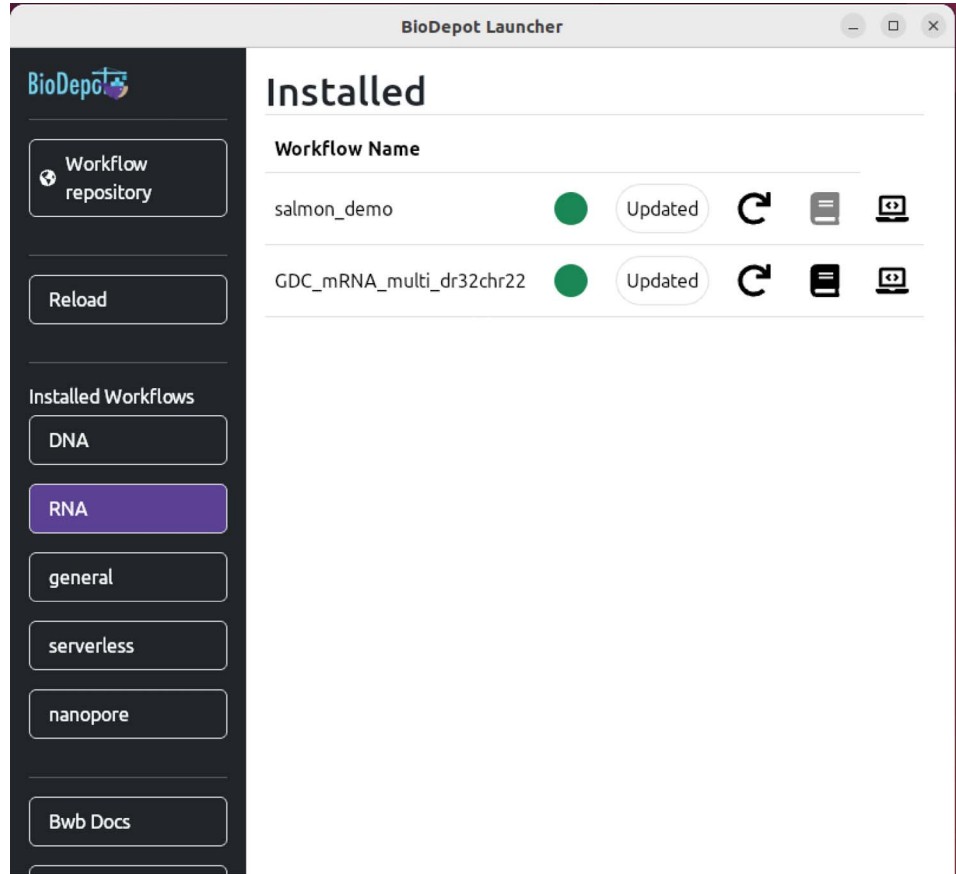

**Figure 3.** A screenshot of the "Installed workflows" screen. In this example, installed "RNA" (RNA sequencing) Workflows are shown. Along side the workflow name of each installed workflow are buttons and indicators. The colored dot, shown here in green, represents the status of the installed workflow. The status button provides a textual status of the workflow (in the case of "Updated"). The circular arrow icon will rebase the workflow. The book icon is a link to the documentation for the workflow on GitHub. If this button is grayed out, there is no documentation. The laptop icon is used for launching the workflow in a variety of ways, including locally and on AWS.

documentation icon opens the documentation in a new browser window. If the documentation is not available on GitHub, the documentation icon will be grayed out.

## Deploying Bwb locally

Another icon (a small computer) present for each workflow is the launcher options icon. Selecting this icon brings up several options: launch Bwb in the "Browser", on "GitPod", and on "AWS", as shown in Figure 4. Selecting the "Browser" button opens Bwb in a new browser window on the local machine.

## Deploying Bwb on the cloud

The "GitPod" launch option does not require any additional dependencies besides Biodepot Launcher. There are two types of GitPod instances: Small and Large. They are differentiated based on the amount of storage and computational resources available. Three of the four workflows only need a Small instance, but a DNA sequencing workflow using cancer

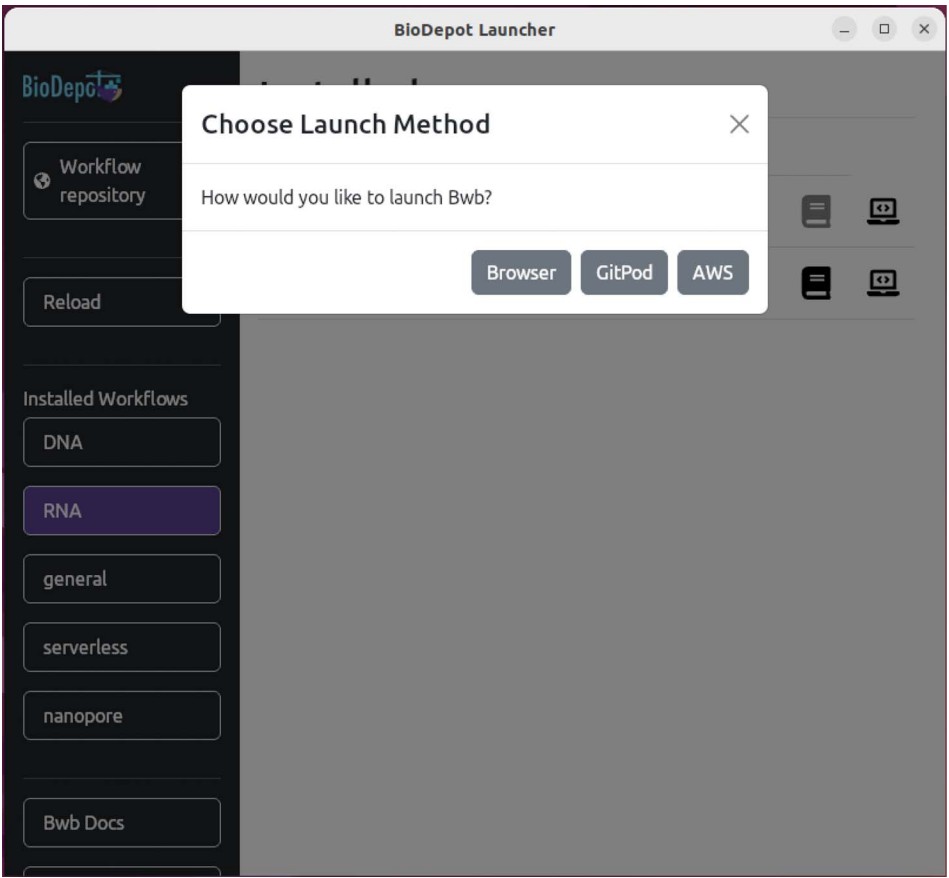

**Figure 4.** Biodepot Launcher can deploy Bwb locally or on the cloud. The "Browser" button opens Bwb in a new browser window on the local machine. The "GitPod" button opens Bwb in a browser on a GitPod cloud instance. And the "AWS" button opens Bwb in a browser on an EC2 instance.

genomics data (GDC-dna-seq-no-gen3-slim-fast) workflow requires a Large instance. Each demo workflow can be run in under half an hour. Notably, the salmon-demo and focal-adhesion-segmentation workflows will run in under five minutes. Additional instructions for how to use GitPod can be found by scrolling-down the sidebar and selecting the "GitPod Docs" button.

AWS CLI and Docker Machine need to be installed to launch Bwb on AWS. AWS CLI should be configured with a working Access ID and Secret Key. Launching an EC2 instance will incur charges to the account specified by the configured credentials. Selecting the "AWS" button will open an additional pop-up that allows the user to enter a different instance type (see Figure 5). Currently, deploying an EC2 instance on AWS requires using region us-east-2 (Ohio).

## CONCLUSIONS

Biodepot Launcher is an open-source application that facilitates the installation and management of workflows, as well as launching Bwb locally, on GitPod or AWS. The launcher contains a curated list of workflows. Use of the launcher helps facilitate the use and adoption of Bwb for a broad range of bioinformatics workflows. The Bwb is a different

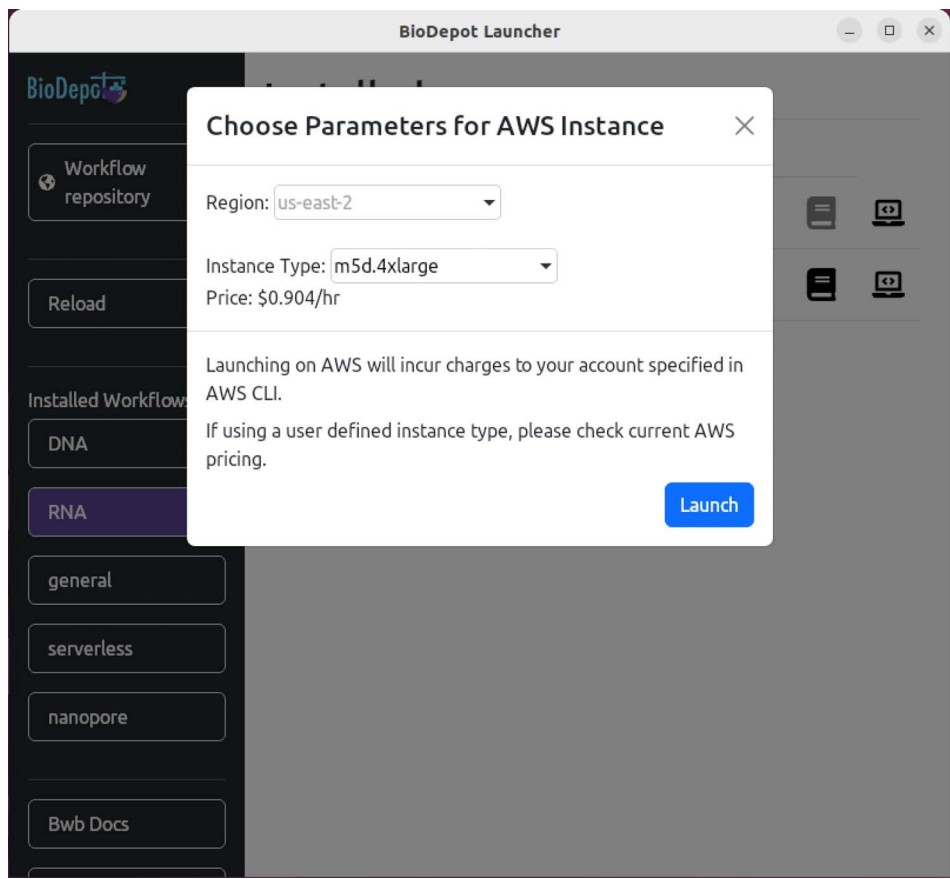

**Figure 5.** When selecting the "AWS" option to deploy Bwb, this parameter selection pop-up will display. The user can specify a different instance type by manually entering it into the text box.

platform that allows the creation and execution of workflows. The Biodepot training portal available at https://biodepot.github.io/training/ includes demo videos and screenshots for the use of the Launcher app which is the focus of this paper. This training portal also includes instructions and details on how to use Bwb to create, customize and execute workflows.

## AVAILABILITY OF SOURCE CODE AND REQUIREMENTS

- Project name: Biodepot Launcher
- Project home page: https://github.com/BioDepot/biodepot-launcher
- Training portal: https://biodepot.github.io/training/running_bwb/bwb_launcher/
- Demo video: https://www.youtube.com/watch?v=8EeKYQkQxxU
- Operating system(s): Ubuntu Linux, Windows 10/11 WSL2 Ubuntu, macOS (M-Series)
- Programming language: Javascript, React
- Other requirements: Docker, Bwb, launcher-utils, AWS CLI (Optional), Docker Machine (Optional)
- License: MIT
- biotools ID: biotools:biodepot_launcher
- RRID:SCR_02613.

## DATA AVAILABILITY

Biodepot Launcher is archived in the Software Heritage repository [18].

## LIST OF ABBREVIATIONS

AWS: Amazon Web Services; Bwb: Biodepot-workflow-builder; CLI: Command Line Interface; RNA-seq: RNA sequencing; SRA: Sequence Read Archive.

## DECLARATIONS

### Ethical approval

Not applicable.

### Competing interests

LHH and KYY have equity interest in Biodepot LLC, which receives compensation from NCI SBIR contract numbers 75N91020C00009 and 75N91021C00022. The terms of this arrangement have been reviewed and approved by the University of Washington in accordance with its policies governing outside work and financial conflicts of interest in research.

### Authors' contributions

LHH and KYY conceived and supervised the study, and contributed to the design of the software application. TJD, LHH, and KYY wrote the manuscript. TJD, EM, RS, and LHH developed the Biodepot Launcher application. JG and TJD wrote the installation scripts. TJD wrote the test plan. LHH, TJD, JG and KYY contributed to the testing of the Biodepot Launcher. All authors read and approved the final manuscript.

### Funding

LHH, TJD and KYY are supported by the National Institutes of Health (NIH) grants U24HG012674 and R03AI159286. LHH and KYY are also supported by NIH grant R21CA280520. LHH and TJD are also supported by NIH grant 3R21CA280520-01S1. LHH, EM, and KYY were also supported by NCI SBIR contract 75N91021C00022.

### Acknowledgements

We would like to thank Jesse Flores, Andrew Jang, Varun Mittal, Niharika Nasam and David Woolston for testing the Biodepot Launcher application across different operating systems. Bryce Fukuda shared the RNA-seq workflows.

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
