## [Reviewer Report]

Indicate in the comments box below whether you are happy with the changes made or if the manuscript is unacceptable.Comments on revised manuscriptI am very happy with the revision. There are a few typos that I hope the journal will correct before publication.Indicate in the comments box below whether you are happy with the changes made or if the manuscript is unacceptable.Comments on revised manuscriptI am very happy with the revision. There are a few typos that I hope the journal will correct before publication.

---

## [Editor Report]

Editor’s AssessmentThis work presents Biodepot Launcher (https://github.com/BioDepot/biodepot-launcher), an open-source, containerized desktop application which facilitates the installation and management of bioinformatics workflows. As well as launching a separate platform called the Biodepot-workflow-builder (Bwb) that is a workflow execution engine. The addition of the launcher providing an end-to-end graphical application that provides a reproducible graphical desktop to create workflows to analyze biomedical data. Alongside containing a curated list of computational workflows. The tool has extensive documentation and training videos, and peer review provided feedback to improve this and the organisation in GitHub. Ultimately this work may be helpful to others that want to create launcher and management apps using a web development framework and libraries such as neutralino.Editor’s AssessmentThis work presents Biodepot Launcher (https://github.com/BioDepot/biodepot-launcher), an open-source, containerized desktop application which facilitates the installation and management of bioinformatics workflows. As well as launching a separate platform called the Biodepot-workflow-builder (Bwb) that is a workflow execution engine. The addition of the launcher providing an end-to-end graphical application that provides a reproducible graphical desktop to create workflows to analyze biomedical data. Alongside containing a curated list of computational workflows. The tool has extensive documentation and training videos, and peer review provided feedback to improve this and the organisation in GitHub. Ultimately this work may be helpful to others that want to create launcher and management apps using a web development framework and libraries such as neutralino.

---

## [Reviewer Report]

Reviewer name and names of any other individual's who aided in reviewerMark A. Jensen, PhDDo you understand and agree to our policy of having open and named reviews, and having your review included with the published manuscript. (If no, please inform the editor that you cannot review this manuscript.)YesIs the language of sufficient quality?YesPlease add additional comments on language quality to clarify if neededOverall, I think the authors can say what they want to say in 30-50% fewer words. Style - is too conversational in some places - suggest being more neutral, brief, and precise.Is there a clear statement of need explaining what problems the software is designed to solve and who the target audience is? YesAdditional CommentsIs the source code available, and has an appropriate Open Source Initiative license <a href="https://opensource.org/licenses" target="_blank">(https://opensource.org/licenses)</a> been assigned to the code?YesAdditional CommentsAs Open Source Software are there guidelines on how to contribute, report issues or seek support on the code?NoAdditional CommentsThere is no CONTRIBUTING.md or similar in the GitHub project repo.Is the code executable?YesAdditional CommentsI executed a few of the tests in the test plan mentioned in the MS, on a 2023 MacBook Pro running Sonoma 14.6.1. Here are my comments: 4. Test: Rebase a workflow. - I added a new folder to the salmon_demo folder, but app did not display “Rebase” button. Continued to display workflow as current. 5. Test: Launch a workflow locally. - the steps succeeded, but BWB opens up with windows stacked on top of each other, it is not clear if the workflow ran or must be executed. 8. Test: Launch a workflow on AWS - the popup disappeared, but no error message or other feedback displayed. No machine was launched in my AWS account. 1. later discovered “error setting machine configuration from flags provided: amazonec2 driver requires AWS credentials configured with the --amazonec2-access-key and --amazonec2-secret-key options, environment variables, ~/.aws/credentials, or an instance role” in dm-output.log file. This should probably be displayed to user.Is installation/deployment sufficiently outlined in the paper and documentation, and does it proceed as outlined?YesAdditional CommentsWalkthrough dialogs and dep installs on initial launch are very helpful. Mac users will see an "Unidentified developer" warning when they double click the app, and it will not run. This can be overridden but I expect not everyone knows how - which is to right-click the app, and select Open from the dropdown. Then the warning will appear in a dialog, but an option to run anyway will appear. When this is selected, the app will run and will be double-clickable from that first run on. It would be worth giving the user the heads up on this in the documentation. Having an Apple Developer account will give access to tools that certify the binary, so that the warning can be avoided altogether.Is the documentation provided clear and user friendly?YesAdditional CommentsIs there enough clear information in the documentation to install, run and test this tool, including information on where to seek help if required?YesAdditional CommentsI would say that, if someone is coming first to the launcher and is unfamiliar with BWB, they will get confused when BWB starts. So some description of BWB would be helpful to include in the launcher docs.Is there a clearly-stated list of dependencies, and is the core functionality of the software documented to a satisfactory level?NoAdditional CommentsThe list of dependencies I believe should include all the external tools that are mentioned in the paper: - include the BWB dependency - include the optional AWS tools dependency - include the Docker engine dependencyHave any claims of performance been sufficiently tested and compared to other commonly-used packages? YesAdditional CommentsIs test data available, either included with the submission or openly available via cited third party sources (e.g. accession numbers, data DOIs)?YesAdditional CommentsMore or less. I believe the paper and the README should describe a Quick Start example based on the test plan. There are a lot of moving parts, so an example that would (or should) run from start to finish would be helpful for the naive user.Are there (ideally real world) examples demonstrating use of the software? YesAdditional CommentsThese are noted in the paper's references.Is automated testing used or are there manual steps described so that the functionality of the software can be verified?YesAdditional CommentsSee above comment on taking the internal test plan and using as basis for a "Quick Start"Any Additional Overall Comments to the AuthorHere are my own notes, hope they are helpful. Overall, same thing can be said in 30-50% fewer words. Thinking of getting the paper read. Style - is conversational in some places - suggest being more neutral, brief, and precise. See below “paying for an EC2 instance…” “Quick start” with example data and workflow? Could use the test plan. Fig.1 caption: “that are ran” -> “that are run” Cost estimation services on AWS that could be employed (maybe in BWB)? Cost limiting within BWB, displayed in Launcher? Does npm or React deserve that much space in the methods? These are very commonly used. Neutralino.js - “creating applications” -> creating desktop applications - this description (p.3) is obscure - instead of “system level”, would say “access to local filesystem, network, and other resources, like a typical desktop application” - “operating environments” -> “operating systems and platforms” Github Actions (p.3) - “Originally…”, are the initial iterations relevant in this paper that introduces the “final product”? I think the motivation of in the first sentence can be deleted. - “This can mean many things…” - delete remarks like this, doesn’t add to the exposition. - Write more tersely and less conversationally. Reduce this paragraph by 50% Docker (p.3) - Is all the detail necessary? PAX format - an ancient archive format. The key point, I think, is that the hash is stable and can be used as a persistent unique identifier on any instance of the archive. Testing (p.3) - “executed by a group of individuals” - meaning that each person independently ran the test plan and reported back? I assume so, but can clarify. Installation (p.4) - First paragraph - would be easier for reader to follow and do if this were in a bulleted format - Instead of “first asterisk, second asterisk” why not write out the convention “neutralino-<linux|mac>-<architecture>.zip”? - “Ubuntu [must] restart” - or can a user run source on an installed script? (i.e., it is just an environment variable change update?) Starting screen (p.4) - Why describe the Start screen in many words, when there is already a screenshot? - “Biodepot launcher is not meant to be left open” - Is it really a problem to leave the app open, as long as you can reload? I would think that a dev would want to make workflow changes and then conveniently reload and rerun. Options to start workflows (p.3) - “but paying for an EC2 instance” - clarify to, e.g., “using the AWS option will enable the user to select appropriate resources; however, the user will incur usage costs.” Deploying on the Cloud (p. 5) - Include AWS cli requirement as optional in the Availability of source code section - I would note for the reader that using AWS requires a access ID and secret key for their own account, and will incur costs. Availbility of source code (p.5) - include the BWB dependency - include the optional AWS tools dependency - include the Docker engine dependency I think launcher should create its own named subdirectory to store the workflow directories and other items. These many items were installed individually in my home directory and clutter it. The names of these items (“RNA”) are generic and might already exist in a user’s home directory.RecommendationMinor Revisions

---

## [Reviewer Report]

Reviewer name and names of any other individual's who aided in reviewerYoann DufresneDo you understand and agree to our policy of having open and named reviews, and having your review included with the published manuscript. (If no, please inform the editor that you cannot review this manuscript.)YesIs the language of sufficient quality?YesPlease add additional comments on language quality to clarify if neededIs there a clear statement of need explaining what problems the software is designed to solve and who the target audience is? YesAdditional CommentsThe goal is clear but there is no comparison with other similar tools. For example, what are the advantages and inconvenient regarding Galaxy ?Is the source code available, and has an appropriate Open Source Initiative license <a href="https://opensource.org/licenses" target="_blank">(https://opensource.org/licenses)</a> been assigned to the code?YesAdditional CommentsAs Open Source Software are there guidelines on how to contribute, report issues or seek support on the code?NoAdditional CommentsNot clear what developers can do to help. There is a debug command line but no guidelines for the future of the software.Is the code executable?YesAdditional CommentsIs installation/deployment sufficiently outlined in the paper and documentation, and does it proceed as outlined?YesAdditional CommentsIs the documentation provided clear and user friendly?NoAdditional CommentsThere are guidelines on how to install but no user manual available from the github repoIs there enough clear information in the documentation to install, run and test this tool, including information on where to seek help if required?YesAdditional CommentsIs there a clearly-stated list of dependencies, and is the core functionality of the software documented to a satisfactory level?NoAdditional CommentsHave any claims of performance been sufficiently tested and compared to other commonly-used packages? Not applicableAdditional CommentsIs test data available, either included with the submission or openly available via cited third party sources (e.g. accession numbers, data DOIs)?NoAdditional CommentsAre there (ideally real world) examples demonstrating use of the software? NoAdditional CommentsIs automated testing used or are there manual steps described so that the functionality of the software can be verified?NoAdditional CommentsAny Additional Overall Comments to the AuthorI think that the authors should focus on extensive documentation for users. In that documentation they should explicit : - How to use a pipeline and focus on 1 example - How to include a new pipeline - How to contribute in general to the project For now there is only documentation for specialist users on how to install the tool.RecommendationReject (Unsound or Unusable)